# Factors Associated with Burnout in Medical Staff: A Look Back at the Role of the COVID-19 Pandemic

**DOI:** 10.3390/healthcare11182533

**Published:** 2023-09-13

**Authors:** Sabinne-Marie Țăranu, Ramona Ștefăniu, Tudor-Ștefan Rotaru, Ana-Maria Turcu, Anca Iuliana Pîslaru, Ioana Alexandra Sandu, Anna Marie Herghelegiu, Gabriel Ioan Prada, Ioana Dana Alexa, Adina Carmen Ilie

**Affiliations:** 1Department of Medical Specialties II, Faculty of Medicine, Grigore T. Popa University of Medicine and Pharmacy, 700115 Iași, Romania; 2Department of Geriatrics and Gerontology, Faculty of Medicine, Carol Davila University of Medicine and Pharmacy, 020956 București, Romania

**Keywords:** burnout, COVID-19, medical staff, non-pandemic stressors, pandemic stressors

## Abstract

Despite the significant consequences for medical practice and public health, burnout in healthcare workers remains underestimated. Pandemic periods have increased the reactivity to stress by favoring some changes whose influence are still felt. Purpose: This study aims to identify opportune factors during pandemic periods that predispose medical personnel to burnout and the differences between medical staff which worked with COVID-19 patients and those who did not work with COVID-19 patients. Material and Methods: This is a prospective study on 199 subjects, medical staff and auxiliary staff from national health units, COVID-19 and non-COVID-19, who answered questions using the Google Forms platform about the level of stress related to the workplace and the changes produced there. All statistical analyses were conducted using IBM SPSS Statistics (Version 28). Results: The limited equipment and disinfectant solutions from the lack of medical resources category, the fear of contracting or transmitting the infection from the fears in relation to the COVID-19 pandemic category and the lack of personal and system-level experience in combating the infection due to the lack of information on and experience with COVID-19 were the most predisposing factors for burnout. No significant differences were recorded between those on the front line and the other healthcare representatives. Conclusions: The results of this study identify the stressors generated in the pandemic context with prognostic value in the development of burnout among medical personnel. At the same time, our data draw attention to the cynicism or false-optimism stage of burnout, which can mask a real decline.

## 1. Introduction

Burnout syndrome is a current issue for healthcare workers. It is characterized by an insidious pathology with a diagnosis prone to confusion, and it is recognized as a challenge for all researchers who have studied it over time. Simply defined, burnout is a psychological syndrome with symptoms of psycho-emotional and physical exhaustion and emotional and work performance loss, triggered by the exaggerated increase in tasks and the decrease in individual coping resources. Since 1970, when psychoanalyst Freudenberger introduced the concept, the legitimization of burnout has varied. First, it was defined as the imbalance between increased workloads and reduced reserves for their fulfillment. Later, it was understood by using three defining dimensions: exhaustion, cynicism and the loss of efficiency at work [1,2,3,4,5].

Although long studied, conceptualized and redefined, burnout remains an underestimated pathological entity. The proof of this fact is the continuously increasing rate of its complications, affecting medical staff, the medical approach and patients. There is still an inclination to superficially understand burnout in order to minimize such a complex attack on the homeostasis of psychic and physical defense resources against stress, which leads to the onset of a cumulative and subconscious decline in the conservation instinct. Therefore, the lack of responsibility in recognizing and nosographically understanding burnout remains a public health issue.

In the medical field, staff face challenges of a physical (insufficient time to observe a schedule of food, hydration or rest), emotional (empathizing with the problems of patients and their families), cognitive (the constant need to update and memorize medical information) and moral nature (ethical or bureaucratic compromises). These contexts, added to the individual degree of approach and resilience, lead, most frequently, to the installation of burnout [6,7,8,9]. The repercussions on the quality of a medical act are what make the difference between burnout and other fields of activity by decreasing the professionalism of the staff, with the possibility of serious medical errors and loss of trust in the medical system [10,11,12].

The reported incidence of burnout varies worldwide. However, the medical personnel most affected by burnout are doctors. Shanafelt et al. observed an increased incidence of burnout symptoms in US physicians (37.9%) compared to the control population (27.8%) [13]. The incidence of burnout varies both in the European Union and in countries outside the European Union. It is lower in European Union countries (10%) compared to countries outside the European Union (17%). Thus, Slovenia seems 5 times more affected (20.6%) than Finland (4.3%), while Turkey is almost 2 times more affected by burnout (25%) than Albania (13%) [14]. The quality of life at the national level, the remuneration, the degree of education, the respect of society and also the climate can be factors that can be individually correlated with the onset of the burnout syndrome [14,15].

Numerous studies so far have demonstrated the permanence of certain factors, both associated with the medical field of activity and independent of it, which maintain the mental and physical reactivity to stress [6,7,8,9,10,11,12,13,14,15,16].

There are numerous stressful factors generated by the work environment that are correlated with burnout in medical staff. Some examples are as follows: increased demands from superiors, the number of hours overtime, low experience in the activity, lack of autonomy in making decisions at work, conflict situations at work, interaction with depressed/violent patients, medical errors, increased responsibility for decisions, reduced career opportunities, insufficient remuneration, defective functionality of the organization, exaggerated bureaucracy, the detriment of patients, etc. [10,11,12,13,14,15,16].

The current literature identifies numerous predisposing factors for burnout in medical personnel. Thus, a meta-analysis classified these stressors as being related to the workplace (e.g., medical specialty and professional degree) and independent of it (e.g., demographic aspects and associated comorbidities). Moreover, female sex, young age and concern for patients are the most common stressors among medical personnel that are associated with burnout. These stressors are predisposing factors for burnout and are associated with a higher degree of psycho-emotional vulnerability [8].

The pandemic period enhanced these stressors mentioned above and this pandemic became a factor of increased a priori stress level. Data showed that the burnout rate increased during the 2005 and 2011 epidemics, returning, post-epidemiologically, to the previous stage [17,18].

There are numerous stressors generated by crisis contexts, such as pandemics, that can explain the onset of burnout syndrome: the lack of protective equipment, the reorganization of intra-hospital circuits, the fear of infection and transmitting the infection, the need to make difficult ethical choices regarding the prioritization of care, family isolation, the experience of quarantine, the feeling of insecurity related to the ability of the work environment to control and fight the infection, the deterioration of intercollegiate relations related to the distribution of spores and the degree of exposure to infection, specialties with a high degree of exposure (ATI, UPU and pneumology), the lack of sufficient medical information about the complications and how to combat the new disease, the lack of prognostic tools, prolonged use of protective equipment, social distancing, much more frequent contact with suffering and death, stigmatization by society and relatives, extensive media alerts, coexisting chronic pathologies and specific tests to detect COVID-19, performed repeatedly [17,18,19,20,21,22,23].

Many results in the literature have revealed an increased prevalence of fear of infection, anxiety, depression, post-traumatic stress and sleep disorders in medical personnel, the latter having the highest rate during pandemics [16,17,18]. Another meta-analysis showed that the rate of depression is higher in healthcare workers who had a higher degree of exposure to the disease, both during the COVID-19 pandemic and during other SARS pandemics [19].

Psychological interventions are recommended in the early stages of burnout. Their advantages are represented by the improvement in the quality of personal and professional life. The disadvantages are represented by the worsening of symptoms, the triggering of burnout and the decrease in efficiency at work [19]. Recommendations are theorized for the support of medical personnel during pandemics through interventions both in the personal sphere and at the level of the medical system. Most studies focus on psychological support [20,21,22]. Individual resilience, however, plays a fundamental role in managing stress reactivity [22].

However, recent data highlight the adverse effects of exhaustion, in a pandemic context, in the entire area of medical activity: the decrease in the desire for progress, professional satisfaction, the quality of medical services and implicitly, the decrease in the level of confidence and comfort of patients, with an increase in the number of infections related to medical assistance and mortality among patients. Coinciding with this vicious circle is the tendency of a significant percentage of the medical staff to resign [23,24,25].

There are current directions on how to prevent burnout syndrome at the medical team level. These include the optimization of the staff schedule, optimization of the number of patients taken by each staff member, improvement in protective equipment and operating circuits and provision of special transport and accommodation in the necessary circumstances [26]. Another recent study refers to an online platform for educating and counseling medical staff (Battle Buddy), with an interest in specific steps for launching a psychological resilience intervention [27].

Burnout not only affects an individual’s quality of life, but also the quality of a medical act and the costs of medical care. General and individualized interventions to prevent burnout and its identification in the early stages remain fundamental principles for combating it. Thus, knowing those stressful factors, both related to the work environment and independent of it, represents a method of preventing burnout. Also, in recent cases in which burnout has installed, recognizing and eradicating these factors could be a way of healing.

Although current data look at burnout through its predisposing factors, we must be aware that these, in certain contexts, can be reconfigured and that new stressors can always appear. That is why, in the medical field, the aspects associated with burnout always remain open to research.

Burnout syndrome was an important problem for healthcare personnel even before the pandemic. Increased workloads, the number of hours worked overtime, the bureaucratic pressure and pressure from management, the emigration of residents to other countries and the shortage of staff, insufficient remuneration and the difficulty of career advancement are just some of the general stress factors for burnout in medical workers before COVID-19. To these factors are added more, such as the decrease in professional satisfaction and self-blame, maintained by working with patients with multiple comorbidities or in the palliation stage, with the gradual installation of emotional and physical exhaustion and later even with the loss of professionalism. Even before the pandemic, more and more healthcare personnel were faced with burnout syndrome. This fact was treated superficially, due to the difficulty of starting collective methods and programs to combat and increase individual and collective resilience. Moreover, this aspect has always been a vulnerable point, since substantial educational and financial resources are needed to implement these programs to combat burnout. Understanding the seriousness of the problem and the recognition of the stressors, especially those that are modifiable, can constitute minimal methods of preventing burnout [23,24,25,26,27].

Despite the fact that all the relevant literature legitimizes burnout syndrome with its maintenance factors, the recourse of these data needs strengthening. In the medical field, the constant reshaping of stress conditions requires a permanent update, and the aspects associated with burnout always remain open to research.

The aim of our study is to identify the stressors associated with burnout syndrome, both generated by a crisis context such as the pandemic as well as general, non-pandemic ones, and to observe differences between medical staff that worked with COVID-19 patients and those that did not work with COVID-19 patients. 

## 2. Material and Methods

### 2.1. Study Design and Setting

The study is a prospective population-based cohort study using a descriptive correlational design. We performed a prospective study on a group consisting of 199 subjects from the medical field. The study was conducted in accordance with the Declaration of Helsinki and approved by the Ethics Committee of Parhon Hospital, Iasi (Approval Code: 9891). The questionnaire was distributed and answered between September 2021 and March 2022. The study participants responded both during the different pandemic waves and during the inter-critical periods. However, the information on the ways to spread and prevent the disease was much more up to date in later waves than in the first wave. 

### 2.2. Study Population

The study population consisted of medical professionals (doctors, resident doctors, nurses and orderlies) and auxiliary staff from COVID-19 and non-COVID-19 national health facilities in Romania. Our subjects freely consented to answer questions and they chose whether to remain anonymous or provide us their emails [28]. 

#### 2.2.1. Inclusion Criteria

The inclusion criteria were age over 18 years, working in the health field (doctor, resident doctor, nurse, orderly or auxiliary staff), working in Romania and the presence of informed consent to participate in the study.

#### 2.2.2. Exclusion Criteria

The exclusion criteria were age < 18 years, work in a field other than healthcare, work in a country other than Romania and absence of informed consent to participate in the study.

### 2.3. Data Collection and Measures

#### 2.3.1. Procedure

The questionnaire was created using the Google Forms platform and distributed through professional groups on online social media platforms (WhatsApp and Facebook) or was disseminated using professional email addresses in our hospital database. Thus, the online dissemination of the questionnaire helped to spread it to medical offices and COVID-19/non-COVID-19 hospitals throughout the entire country (22 different counties) while maintaining the anonymity of the participants. 

The objectives of this research were to identify the factors associated with burnout syndrome, to determine their statistical significance and to identify differences between medical staff that worked with COVID-19 patients and those who did not work with COVID-19 patients.

#### 2.3.2. The Screening Tool

We created our questionnaire using both original questions developed by our team and questions from the Maslach Burnout Inventory as well as from a burnout evaluation questionnaire published in 1993 by Katrina Shields in *In the Tiger’s Mouth: An Empowerment Guide for Social Action* [3,4]. We organized the screening tool in 2 sections as follows. In the first section, we included 105 original questions related to the activity of the participants: questions 1–44 encode variables from the literature associated with burnout among medical personnel, including categories of stressors like poor workplace quality of life, personal life implications, exaggerated bureaucratic work, ethical issues and self-culpability and career progress issues; questions 45–105 encode variables from the literature associated with the burnout of medical staff in the context of the pandemic, including categories of stressors like lack of medical resources, poor workplace quality of life, worsening of personal life quality, limiting of free time, fears in relation to the COVID-19 pandemic, work environment reorganization, lack of information on and experience with COVID-19 and worsening of medical care quality.

In the second section we included 2 burnout evaluation questionnaires from the literature. Questions 1–16 were represented by the Maslach Burnout Inventory, for which we obtained an authorized translation [3]. Questions 17–26 were represented by the burnout evaluation questionnaire, a version adapted from the original one published in 1993 by Katrina Shields in *In the Tiger’s Mouth: An Empowerment Guide for Social Action*, for which we obtained the author’s consent to use [4].

The total burnout score was calculated using the second section from our screening tool, represented by 26 questions. All 26 items on the burnout scale were answered on a six-point scale ranging from 1, strongly disagree, to 6, strongly agree. This scale was chosen to avoid the tendency of subjects to choose a “neutral” middle response. There were no inverted items that required recoding of the variable. This means that for all items, a lower score indicates a lower level of burnout, and a higher score indicates a higher level of burnout.

Internal consistency was found to be excellent for the original set of items. Because the original scale used a total score in tandem with separate scores for the factors, we summed the score for each item and calculated a total burnout score for each participant. Despite the excellent internal consistency for the entire scale, the total burnout score was not normally distributed across participants (One-Sample Kolmogorov–Smirnov Test Z = 1.59; *p* < 0.05). 

For our entire questionnaire, the Cronbach’s alpha coefficient was determined at 0.97, an optimal value.

### 2.4. Data Analysis

All statistical analyses were conducted using IBM SPSS Statistics (Version 28). For continuous variables, the Mann–Whitney U test was used, with mean and interquartile ranges reported as descriptive statistics, along with the strength of the effect, r. Categorical variables were compared using the chi-square test. The Kolmogorov–Smirnov normality test indicated significant variations in the total burnout scores in relation to the normal curve. In the quantitative interpretation of the data, we used Kruskal–Wallis and Mann–Whitney analysis. We used Student’s *t* test to compare 2 groups for normal distributed parameters and Mann–Whitney analysis for the rest. Statistical significance was defined at *p* = 0.05. 

## 3. Results

The total study population consisted of 199 healthcare representatives.
A. The demographic data of our group are presented in Table 1.

Our group was represented by 199 participants from the medical field, of which 167 (83.91%) worked with COVID-19 patients and 32 (16%) did not work with positive patients. The two subgroups were compared according to their sex, workplace location, profession, work type, workplace seniority, marital status, living conditions and children and seniors in care.

The majority of the study group was represented by medical personnel who worked with positive COVID-19 patients (83.91%). Among them, most were female (89.82%), working in urban areas (91.62%), resident doctors (41.92%), working in a hospital (91.02%), married (55.09%), living in an apartment (70.66%) and without children (56.29%) or elderly (56.29%) in care.

Between those who worked with positive patients and those who were not exposed to this risk, there were statistically significant differences regarding the profession (*p* = 0.045, r = 0.047) and the type of work (*p* = 0.015, r = 0.003) (Table 1).

There were no statistically significant differences between those who worked on the front line and those who did not take care of positive patients regarding sex (*p* = 0.488, r = 0.491), workplace location (*p* = 0.685, r= 0.686), workplace seniority (*p* = 0.480, r = 0.163), marital status (*p* = 0.555, r= 0.525), living conditions (*p* = 0.221, r = 0.222), children in care (*p* = 0.671, r= 0.730) and seniors in care (*p* = 0.111, r = 0.158) (Table 1). Thus, according to our results, the demographic data of the participants do not statistically influence the level of burnout of the two compared groups.

B. Correlations between burn-out level and pandemic/non-pandemic stressors

(1)Degree of exposure

According to our results, for the 167 participants who worked with positive COVID-19 patients, the average level of the burnout score was 99.41. Also, for the 32 participants who did not work with positive COVID-19 patients, the average level of the burnout score was 103.9. (Figure 1).

There are no significant differences between those who worked directly with patients with COVID-19 and those who did not work directly with patients with COVID-19 in terms of the average level of burnout, as shown by the Mann–Whitney statistic (U = 2.57; *p* > 0.05) (Figure 1).

(2)Pandemic stressors associated with burnout in all medical staff and those on the front line

We grouped the pandemic stressors into generic categories such as lack of medical resources, poor workplace quality of life, worsening of personal life quality, limiting of free time, fears in relation to the COVID-19 pandemic, work environment reorganization, lack of information on and experience with COVID-19 and worsening of medical care quality.

According to our data, the factors positively correlated with the total score of burnout among medical personnel are especially from the first category, the lack of medical resources: limited protective equipment (*p* = 0.01, r = 0.358) and disinfectant solutions (*p* = 0.01, r = 0.370) and the need to buy or improvise protective equipment (*p* = 0.01, r = 0.303). Also, we noticed that categories of pandemic stressors such as limiting of free time, work environment reorganization and fears in relation to the COVID-19 pandemic are significantly related to burnout among medical staff. Thus, working overtime (*p* = 0.01, r = 0.407), reorganization of intra-hospital circuits (*p* = 0.01, r = 0.417) and fear of contracting (*p* = 0.01, r = 0.379) and transmitting SARS-CoV-2 infection (*p* = 0.01, r = 0.408) can be considered important pandemic stressors. 

Also, the physical somatization of stress (pharyngeal pain associated with infection) (*p* = 0.01, r = 0.315), quarantine (*p* = 0.01, r = 0.498), the feeling of insecurity regarding one’s experience (*p* = 0.01, r = 0.364) and the system in case management (*p* = 0.01, r = 0.419) are directly related to the total burnout score for all medical personnel (Table 2). 

At the same time, the prioritization of patients according to the vital prognosis (*p* = 0.034, r = 0.150), the feeling of loneliness in the fight against the virus (*p* = 0.017, r = 0.169), the fear of marginalization by those close to them due to the interaction with positive patients (*p* = 0.008, r = 0.187) and the need for repeated self-testing (*p* = 0.018, r = 0.168) were statistically significantly associated with stress reactivity in all healthcare personnel. Paradoxically, for the participants in the first line of fighting the disease, these variables were not significant.

No significant results were reported in relation to total burnout and the low number of days off, the impact on the quality of life, the lack of information about the complications and the way to eradicate the infection, the permanent change in combat protocols and the individual work environment, the prolonged wearing of protective equipment and the adaptation to low material resources in the context of increased workloads (Table 2).

Our results show that only the lack of experience in caring for infected patients correlates positively, both with the total burnout score for those on the front line and for the entire medical staff (*p* = 0.030, r = −0.168). Also, only the increasing number of positive cases was related to stress in those on the front line of fighting the disease (*p* = 0.030, r = −0.168). The rest of the variables generating stress and related to the COVID-19 pandemic mentioned above were not statistically significantly associated with the total burnout score in directly exposed personnel (Table 2).

(3)Total burnout score for those working with COVID-19: nonspecific variables

At the same time, we classified non-pandemic stressors into several categories, such as poor workplace quality of life, personal life implications, exaggerated bureaucratic work, ethical issues and self-culpability and career progress issues.

Among the non-pandemic factors providing an increase in reactivity to stress, only the association of psychiatric conditions under treatment in those who worked in the COVID-19 sectors was positively correlated with the total burnout score (*p* = 0.08, r = 0.204) (Table 3). It is interesting that only the personal life implications category was statistically represented by this single stressor.

Even if the other categories such as poor workplace quality of life, exaggerated bureaucratic work, ethical issues and self-culpability and career progress issues were not statistically representative, they include stressors that can maintain reactivity to stress, although they do not correlate with burnout, according to our data.

## 4. Discussion

Regarding the average level of burnout and the general data of the study group, our research did not identify significant differences between the average level of burnout in the subgroup that worked with positive patients and the one that was not on the front line of fighting the disease.

Thus, in our study, the majority of participants who interacted with COVID-19-positive patients worked in the hospital, being represented mostly by resident doctors. Data from the literature corroborate our evidence, showing an increased predisposition to burnout in resident doctors during the pandemic [29,30,31,32]. However, there are also data that attest to an increase in the level of stress, especially among resident doctors, independently of the influence of the pandemic [33].

Also, our results do not illustrate a statistically significant correlation between the level of burnout and sex or living with other people. We believe that this aspect is also explained by the fact that the majority of respondents cared for patients with COVID-19 (83.91%) and their reporting to the minority in non-COVID-19 sectors decreased the value of the associations. An analysis showed that during the COVID-19 pandemic, anxiety, depression and social stress, which are predictors of burnout syndrome, were positively associated with older age, female sex, degree of exposure to illness, lower number of days off and living with family members or other people [18].

Even if other data from the literature show that female patients, young age and concern for patients are vulnerable categories, our results do not identify these factors as predisposing to burnout [8]. Also, according to our results, the categories of stress factors related to the work environment did not prove statistically significant, although some data recognize their importance in this direction [8]. 

We compared the two subgroups (COVID-19 working, non-COVID-19 working) for the questions on the burnout scale with the aim of identifying the group most affected by burnout from a statistical point of view and at the same time recognizing the most vulnerable profile of those from the field of health in combating the next pandemics.

According to our results, there are no significant differences between those who worked on the front line and those who did not work directly with patients with COVID-19 in terms of the average level of burnout. We think that this result is due to the fact that most responses were collected during periods of declining caseloads when stress reactivity became refractory, and anxiety and insecurity were replaced by skepticism. However, most data in the literature show higher rates of burnout in front-line healthcare workers [34,35,36,37,38,39,40,41].

We felt the need to group stressors, both pandemic and non-pandemic, into larger categories, since even those that did not prove statistically significant for the onset of burnout can potentiate the stressful effect of others in the same category. In addition, these categories designate general circumstances precipitating burnout.

Therefore, we classified the pandemic stressors into generic categories, such as lack of medical resources, poor workplace quality of life, worsening of personal life quality, limiting of free time, fears in relation to the COVID-19 pandemic, work environment reorganization, lack of information on and experience with COVID-19 and worsening of medical care quality.

We also grouped non-pandemic stressors into several categories, such as poor workplace quality of life, personal life implications, exaggerated bureaucratic work, ethical issues and self-culpability and career progress issues. 

Our data support that the fears in relation to the COVID-19 pandemic category represented by the fear of transmission and self-infection correlate positively with stress reactivity in all medical staff. However, according to these results, burnout syndrome is not statistically significantly associated with these variables in those on the front line. We consider that we obtained these results due to the fact that most of the data were collected in the inter-critical periods, when burnout reached a decline in cynicism or self-imposed optimism. 

One of the most important pandemic stressors is the fear of transmitting the infection to family and contracting the disease. Thus, a study shows that the prevalence of psychosomatic and psychological symptoms in front-line medical workers during the COVID-19 pandemic was predominantly represented by this factor [42]. According to other results, the fear of infection was also positively associated with burnout and its predictors (depression, anxiety and stress scores), and doctors and nurses who worked in direct contact with infected patients had higher rates of burnout, depressed mood, job stress and job dissatisfaction compared to those less exposed [43]. In this sense, our results confirm the importance of the feeling of fear regarding contracting and transmitting the infection in triggering burnout syndrome.

Also, another study demonstrated that the impact of the COVID-19 pandemic was statistically significant in terms of concern about transmitting the disease and awareness of its impact on society. Paradoxically, the same data suggest that sleep disturbances, suicidality and nightmares, as consequences of burnout, did not worsen during the pandemic. However, the frequency of burnout syndrome increased significantly during this time [21]. The very impact of the outbreak of an emerging disease has been related to the installation of a general atmosphere of fear, highlighting the importance of laborious research activity to understand its possible negative effects on the psycho-emotional health and productivity of individuals. We want to mitigate such an impact on healthy people and, in this case, those on the front lines of fighting the disease, as the predisposition to mental vulnerability has been demonstrated, especially in the latter [42,43,44].

At the same time, our results attest that the lack of information on and experience with COVID-19 category represented by a feeling of insecurity regarding one’s experience and the system’s experience in case management was a factor precipitating the level of stress among the entire medical staff. The literature shows that during the periods of mobilization of health representatives to the outbreak areas, the first concrete stressors of the epidemic were recorded: the fear of infection and the feeling of insecurity, also maintained by extensive media alerts [45].

According to our results, we noticed that categories of pandemic stressors such as the lack of medical resources, work environment reorganization and fears in relation to the COVID-19 pandemic are significantly related to burnout among medical staff.

Our results report the lack of disinfectant solutions and protective equipment, with the need to improvise them, as pandemic stressors positively associated with burnout among all medical personnel. Evidence from the time of COVID-19 recalls that the majority of medical personnel showed their solidarity and petitioned to join the action to combat the spread of the disease. However, at that time the anticipations regarding the possible psychological disorders arising in the context of the confrontation with the growing epidemic and the uncertainty regarding the lack of medical material, especially personal protective equipment, were premature [45].

Also, our results show that the reorganization of intra-hospital circuits and the increase in overtime work were predisposing factors for the installation of burnout among the entire medical staff. Data from the literature document the real influence of these variables, especially in those on the front lines of disease control. Thus, the lack of opportunity to rest adequately correlated with extremely high workloads and the lack of rotation programs can lead to staff exhaustion with mental deconditioning in a very short time [20,45]. 

However, the data in the literature focus on the impact of circuit reorganization and healthcare efficiency suffering because of improper patient–medic interactions. In addition, the stress of schedules, modified shifts, prolonged wearing of protective equipment and masks with an unhealthy hermetic seal, and lack of nutrition, hydration and rest are recorded. The process of equipping/unequipping is associated with high risks of autoinfection and infection transmission, itself being a stress factor [40,46].

The literature also refers to the depersonalization of medical acts as a consequence of shortening the time of interaction with patients, but also of the use of protective equipment, with a decrease in the possibility of empathy, socialization and mutual encouragement [47,48,49]. Our study reports the tendency toward the physical somatization of stress in all healthcare representatives, with acute-onset pharyngeal pain being closely associated with the stress of infection. Recent studies have confirmed this aspect among those who worked in the health field during the COVID-19 outbreak [40,48,49]. 

Some data suggest that respiratory and digestive somatization in those assigned to the front line was found to have an increased prevalence compared to those on the other lines, although no significant differences were recorded between coping styles [48,49].

It was also shown that the most experienced symptoms among medical personnel in the last month of the COVID-19 outbreak were psychological ones and predictors of burnout (anxiety, fatigue and insomnia), but also those related to the prolonged wearing of protective masks (neck pain and headache), all with a moderate degree of impairment [48].

Our results also showed that the fear of marginalization by those close to them constituted an important stressor for all medical personnel. However, those on the front line were not affected in this sense. Data from the literature have instead highlighted an increase in stigma among those exposed on the front line, who are often marginalized by colleagues in less-exposed medical fields or by acquaintances based on their consideration as vectors of infection [49].

Lack of experience with caring for infected patients is positively correlated with the total burnout score in both front line and overall healthcare staff, according to our study. Also, consistent with our results, the increasing number of positive cases was related to stress in those on the front lines of fighting the disease. A multicenter survey investigating psychological disorders in medical personnel without any experience in caring for the disease and working in different medical fields showed statistically significant differences between groups, finding a significant increase in total scores of anxiety, depression, sleep quality and stress in participants at high risk of exposure to COVID-19. Further post hoc analysis indicated that staff posted in Hubei, where the number of cases was consistently exponential, had higher anxiety, depression, sleep quality and stress scores [50].

Although our results do not identify significant associations between burnout and pandemic stressors in those who worked with positive patients, the history of epidemics shows an always higher prevalence of stress levels in those exposed to a higher percentage [13,50].

Even if non-pandemic stressors are the most common and recognized categories of general stressors, they were not statistically representative according to our data, but we should remind that they include stressors that can maintain reactivity to stress. 

Interestingly, in agreement with our results among the non-pandemic factors predisposing to burnout, only the association of psychiatric conditions under treatment was positively correlated with the total burnout score in those who worked in the COVID-19 sectors, although some data consider that this category of stressors is much wider [8].

According to the literature, current issues of particular relevance to the global medical world include the links between burnout syndrome and mental illness, with the differential diagnosis being necessary and difficult to perform in certain clinical circumstances [51,52,53,54].

We believe that recognizing stressors or the large categories they belong to means both a method of preventing burnout but also a method of increasing individual resilience through awareness of the importance of eradicating them from everyday life. All collective methods of combating burnout initially require an individual assumption of the seriousness of the problem.

## 5. Conclusions

Looking back at the COVID-19 pandemic, our data study burnout syndrome in close predictability with pandemic stressors such as fear of infection and transmission of infection, lack of information on and experience with caring for a new disease and the physical and emotional overload generated by total change in the workplace. More attention should be paid to burnout stages where cynicism or false optimism can mask the real state of the subjects. 

Any crisis context can generate new stressors. Their permanent updating and recognition of their importance in triggering burnout is a form of combating it, by increasing individual resilience. Thus, burnout and the factors associated with burnout remain subjects always open to research.

### 5.1. Research Limitations

Our study group was represented mostly by resident doctors; thus, it is not fully representative of medical staff in our country. Respondents answered disproportionately according to their profession and specialization, and we cannot say objectively which categories are most affected by burnout.

Another limitation of our study is the fact that the study participants responded to our questionnaire both in the inter-critical periods, between the pandemic waves, and in pandemic waves; this aspect can change the authentic stage of their burnout.

### 5.2. Recommendations

Knowing the stressors born at the emergence of a critical context means developing the right attitudes to approach these pathologies and burnout syndrome. These represent the valorization of professional qualities, the appreciation of work efforts and the psychological and financial support of medical staff. Burnout in medical staff should be determined both preventively and when it is suspected. We should diagnose and prevent burnout in medical personnel, especially because medical personnel are a vulnerable category of workers. When burnout is detected, comprehensive and immediate measures should be taken. We should learn from previous pandemics to prevent and properly treat burnout in medical personnel, especially as another pandemic could come.

## Figures and Tables

**Figure 1 healthcare-11-02533-f001:**
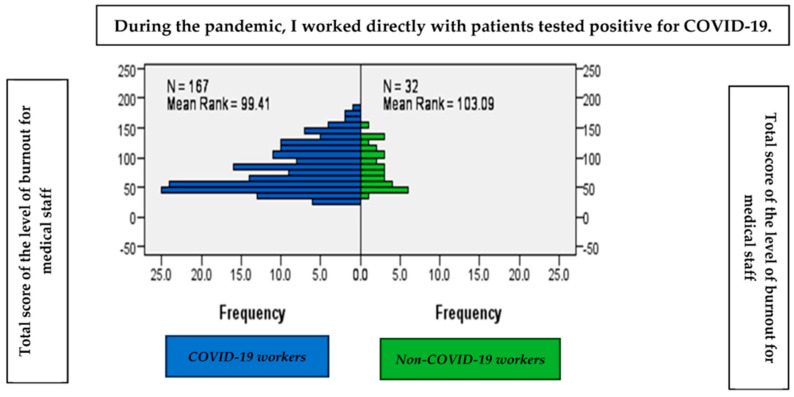
The average level of burnout according to the degree of exposure.

**Table 1 healthcare-11-02533-t001:** General data of the study group.

		COVID-19 Healthcare Staff	Non-COVID-19 Healthcare Staff	*p*	r
**Total workers**		167 (83.91%)	32 (16%)		
Sex	Female	150 (89.82%)	30 (93.75%)	0.488	0.491
	Male	17 (10.18%)	2 (6.25%)		
Workplace location	Rural	14 (8.38%)	2 (6.25%)	0.685	0.686
	Urban	153 (91.62%)	30 (93.75%)		
Profession	Senior doctor	43 (25.75%)	10 (31.25%)	0.045	0.047
	Resident doctor	70 (41.92%)	13 (40.63%)		
	Nurse	47 (28.14%)	6 (18.75%)		
	Orderly	6 (3.59%)	0 (0%)		
	Auxiliary personnel	1 (0.6%)	3 (9.38%)		
Work type	Hospital	152 (91.02%)	23 (71.88%)		
	Ambulatory	9 (5.39%)	7 (21.88%)	0.015	0.003
	Ambulance	6 (3.59%)	2 (6.25%)		
Workplace seniority	≤5 years’ experience	84 (50.3%)	18 (56.25%)		
	>5 years’ experience	83 (49.7%)	14 (43.75%)	0.480	0.163
Marital status	Married	92 (55.09%)	20 (62.5%)		
	Divorced	9 (5.39%)	1 (3.13%)		
	In a relationship	47 (28.14%)	7 (21.88%)	0.555	0.525
	Single	22 (13.17%)	4 (12.5%)		
Living conditions	House	49 (29.34%%)	6 (18.75%)	0.221	0.222
	Apartment	118 (70.66%)	26 (81.25%)		
Children in care	With	73 (43.71%)	13 (40.63%)	0.671	0.730
	Without	94 (56.29%)	19 (59.38%)		
Seniors in care	With	32 (19.16%)	3 (9.38%)	0.111	0.158
	Without	135 (80.84%)	29 (90.63%)		

**Table 2 healthcare-11-02533-t002:** Correlations between pandemic stressors and total burnout score in all medical staff and those working with patients positive for COVID-19.

COVID-19 Generated Stress Factors	All Medical Staff		Working with COVID-19 Medical Staff	
	*p*	r	*p*	r
**Lack of medical resources**				
During the COVID-19 period, protective equipment was limited.	0.01	0.358	0.585	−0.043
During the COVID-19, I had to improvise or buy protective equipment.	0.01	0.303	0.247	−0.090
Lately, I have had difficulty coping with the lack of material resources, in the context of the increased volume of work	0.111	0.113	0.379	0.069
There were periods when sanitizing solutions were limited.	0.01	0.370	0.711	0.029
Lately, I’ve been frustrated because there weren’t enough machines to help infected patients breathe	0.871	−0.012	0.655	−0.035
**Poor workplace quality of life**				
I felt exasperation due to prolonged wearing of protective equipment	0.436	−0.055	0.257	−0.088
Recently, I have endured high temperatures, in the context of using protective equipment	0.194	0.092	0.878	−0.012
Lately I have not been properly hydrated in the context of using protective equipment	0.305	0.073	0.490	0.054
Lately, I haven’t been able to eat at work, due to the COVID-19 restrictions	0.081	−0.124	0.148	−0.112
I had to endure repeated testing for the detection of SARS-CoV-2	0.018	0.168	0.063	0.144
I feel that the last period has affected my quality of life	0.421	0.057	0.413	0.064
Recently, the relationship with work colleagues has deteriorated	0.836	0.015	0.833	−0.016
I felt alone in the fight against COVID-19 while my colleagues were avoiding	0.017	0.169	0.255	0.089
Lately, I have felt extra pressure because I work in a prestigious hospital	0.283	0.076	0.300	0.081
**Worsening of personal life quality**				
During COVID-19, I had to isolate myself from family and friends	0.644	0.033	0.212	−0.097
Lately, I have tended to drink more alcohol than before	0.257	0.081	0.186	0.103
Lately, I have had a tendency to use narcotics	0.958	−0.004	0.688	−0.031
Lately, the relationship with my family has worsened	0.196	0.092	0.118	0.122
Lately, I have had to avoid meetings with relatives and friends	0.750	−0.023	0.758	−0.024
I had to care for close relatives who were infected	0.232	0.085	0.902	0.010
**Limiting of free time**				
Lately, the number of free days available to me has decreased	0.854	−0.013	0.689	−0.031
Lately, I’ve been working a lot of overtime.	0.01	0.407	0.707	0.029
In the last period, I worked extra shifts, to replace colleagues infected with SARS-CoV-2	0.067	0.130	0.468	0.057
Lately, I felt that the system in which I work is willing to sacrifice me according to its own goals	0.714	0.026	0.393	−0.067
**Fears in relation to the COVID-19 pandemic**				
Lately, I have been living with the fear of getting infected with SARS-CoV-2.	0.01	0.379	0.935	0.006
Lately, I was afraid of transmitting the SARS-CoV-2 infection.	0.01	0.408	0.603	−0.041
During the quarantine, I felt insecure about what was to come.	0.01	0.498	0.603	0.041
The fact that I have associated diseases made me feel the fear of contacting the infection	0.459	−0.053	0.922	−0.008
Lately, any pharyngeal pain made me think that I had contacted the SARS-CoV-2 infection.	0.01	0.315	0.790	−0.021
I feel that I have been exposed more than other colleagues to the risk of infection	0.154	0.101	0.481	0.055
I was concerned when I saw the increasing number of COVID-19 cases	0.455	−0.053	0.670	−0.033
I felt fear when I had to take over the shifts of infected colleagues	0.221	0.087	0.963	−0.004
I felt anxiety when close colleagues fell ill	0.088	−0.021	0.194	−0.101
I felt like I had to fend for myself the whole time I was working with positive patients	0.304	0.073	0.551	−0.046
I was scared by the increased incidence of COVID-19 cases in the local area	0.967	−0.003	0.965	0.003
During the time I worked with positive patients, I was afraid of being marginalized	0.008	0.187	0.208	0.084
**Work environment reorganization**				
Lately, I felt affected by the reorganization of intra-hospital circuits.	0.01	0.417	0.253	0.089
I feel uncertain about the effectiveness of infection control measures, given the frequent changes in action protocols	0.582	−0.039	0.989	−0.001
Recently, the environment in which I worked has changed substantially	0.701	−0.027	0.187	−0.103
Recently, I have been exposed to aerosol-generating procedures and invasive resuscitation interventions (tracheostomy, oro-tracheal intubation).	0.083	0.123	0.333	0.075
I worked in quarantine areas for a long time	0.160	0.100	0.983	−0.002
I was affected by the loss of records of old patients, in the context of the restrictions	0.671	0.030	0.857	0.014
I received the appropriate income increase for the degree of exposure to COVID-19 and the work performed	0.101	0.116	0.251	0.089
**Lack of information and experience on COVID-19**				
I feel dissatisfaction that there is not enough information about the complications and total eradication of COVID-19	0.927	−0.007	0.411	−0.064
I had the feeling that the system in which I work is not prepared to fight the infection with SARS-CoV-2.	0.01	0.419	0.628	−0.038
My lack of experience in caring for patients with COVID-19 worried me.	0.01	0.364	0.030	−0.168
I feel uncertainty about how the condition of each patient with COVID-19 will evolve	0.254	−0.081	0.500	−0.053
I felt worried at the thought that there is no treatment to combat COVID-19	0.124	−0.109	0.271	−0.086
I experienced a sense of frustration following the misinformation about COVID-19 spread by the media	0.725	−0.025	0.326	−0.076
**Worsening of medical care quality**				
The relationship with patients has become difficult due to false information spread by the media	0.869	−0.012	0.885	−0.011
During COVID-19, I had to give care primarily to patients who had a better chance of survival	0.034	0.150	0.313	0.079
Recently, I have faced many cases of deaths among patients	0.517	0.046	0.432	−0.061
Lately, I have experienced increased suffering among the sick	0.612	0.036	0.689	−0.031
I had periods when the number of patients with COVID-19 that I was dealing with increased substantially	0.941	−0.005	0.030	−0.168
Lately, I have interacted more than ever with desperate families of patients	0.634	0.034	0.587	−0.042
Lately, communication with patients has been much more difficult, in the context of the use of protective masks	0.723	−0.025	0.336	−0.075

**Table 3 healthcare-11-02533-t003:** Correlations between non-pandemic stressors and total burnout score in those who worked with positive COVID-19 patients.

Non-Pandemic Stressor Factors for Medical Staff Working with COVID-19	*p*	r
**Poor workplace quality of life**		
I have been working with depressed patients for a long time	0.162	−0.109
I feel a lot of pressure from the hospital management	0.694	0.031
I face a routine in the activities carried out at work	0.795	−0.020
I often interact with violent patients	0.214	−0.097
At work I encounter conflicting situations with colleagues	0.552	0.046
I always feel that I have to do many things in a short time	0.749	−0.025
I felt harassed at work	0.936	−0.06
I felt discriminated against because of my ethnicity	0.770	0.023
When making decisions at work I must consult my superiors	0.600	−0.041
I always interact with patients who are uncooperative or question what I tell them	0.263	−0.087
I am always faced with unrealistic expectations from patients	0.865	−0.013
**Personal life implications**		
Patients call me constantly, including at home and even at night	0.870	0.013
I have a social life	0.272	−0.086
In my free time I always think about patients’ cases	0.244	−0.091
I suffer from psychiatric conditions for which I take medication	0.008	0.204
**Exaggerated bureaucratic work**		
I deal more with medical documents than with patients	0.376	−0.069
It bothers me that I have to deal with the administrative side as well, not just the patients	0.946	−0.05
**Ethical issues and self-culpability**		
I am constantly afraid of making a mistake that could cost a patient’s life	0.491	−0.054
I care more about patients than family	0.768	−0.023
I feel the pressure of my responsibility to patients all the time	0.619	−0.039
I am bothered by certain medical mistakes I have made	0.406	−0.065
I feel overwhelmed in certain cases and cannot react professionally	0.112	−0.124
I feel regret for the decisions I have made in the cases of certain patients	0.220	−0.095
I always put patients first	0.676	−0.033
I feel that I am not completely fair in the professional environment, having to make certain compromises	0.408	0.064
The responsibility I have at work is very high also due to the position I hold	0.557	0.046
**Career progress issues**		
I am worried about the future of my chosen profession	0.874	−0.012
I have time to update my knowledge or learn new things in my field	0.828	−0.017
I am worried that I do not have a permanent employment contract	0.557	0.046
The country I live in has respected me for all the dedication and efforts made for patients	0.133	0.117
I felt respected by the hospital management and colleagues for my work and dedication	0.757	−0.024

## Data Availability

Data are available on request from the authors.

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
