# Peer review of "Factors Associated with Burnout in Medical Staff: A Look Back at the Role of the COVID-19 Pandemic"

_healthcare, 2023, doi:10.3390/healthcare11182533_

Round 1
Reviewer 1 Report
Title: Factors associated with burnout in medical staff. The role of the COVID-19 pandemic – a look back.
Reviewer Comments: Purpose of this study to identify the factors that cause burnout in health care professionals during COVID-19 pandemic. In this prospective study 199 subjects were chosen. Authors found that resident doctors reported the highest level of burnout on the other hand nurses reported the least affected. There is a positive correlation were identified between burnout and variables include lack of PPE, disinfectant solutions, fear of transmitting the infection and contacting. Surprisingly there was no significant difference between the front-line professionals and other health representatives.
Strengths:
1. Author did a good job in addressing vast range of stress factors.
2. Sample size and variations in the study group.
3. These kinds of studies will help governments to prepare for the future pandemics.
4. It’s surprising and interesting to see the correlation between non-pandemic stressors and burnout.
Weaknesses:
1. There is no molecular data or experimentally derived data.
2. P values or 0,01 or 0.01. (Comma or Full stop?)
3. Was study included the subjects from the same hospital or different hospital or in the same location or random population from entire country?
4. What is the status of the Covid 19 treatment or awareness or protective measures during this study in the location where this study was conducted. Since this is studied in second wave of covid 19, people would have already prepared for what’s coming.
Author Response
"Please see the attachment."

Reviewer 2 Report
This article applies quantitative methods to explore the role of the COVID-19 pandemic around the factors associated with burnout in medical staff. This manuscript has obvious shortcomings such as irregular titles, lack of thoroughness and rigor in the logic of the argument, unclear research questions and multiple typos.
I. P1. The title. The first letter of each word might be capitalized. In addition, according to the punctuation and presentation, the current title has two subtitles: “the role of the COVID-19 pandemic” and “A look back”. “Factors associated with burnout in medical staff: a look back to the role of the COVID-19 pandemic” seems to be more reasonable.
II, P1. In Abstract, L15-17, “Residents (42%) responded with the highest level of reported burnout, and nurses (26.5%) were the least affected (r = 36.73, p < 0.01)”. There is no description or illustration of this data in the text.
III.P1-3, In Introduction section, the research objective and questions are not clearly stated; the core theme-burnout was not clearly conceptualized or operationally defined. The literature review deviated from the topic-burnout, and failed to provide a focused and systematic discussion of the factors that precipitate burnout. In addition, split between content, relationship to research topic-burnout not specified, it is a bit of a foregone conclusion.
L62-64, L73-78, and L100-105 state the influence of factors outside the field of medical activity on physical and mental stress, the high incidence of psychological problems among medical personnel during epidemics, and some principles and methods of burnout interventions, respectively, but how this relates to burnout is not specified.
IV. P3, In Materials and Methods section, it would be better to explain clearly the sources of the research participants (e.g., which region of the country, how they were recruited), the structure of the measurement tool, the scoring method, and the scoring results. The presentation of the method of analysis is poorly formulated.
There are multiple unscientific and erroneous statements regarding the method of analyzing quantitative data. Because there are multiple types of data and multiple validation purposes within the text, it would be preferable to state the analytical methods, data, and analytical purposes in a corresponding manner.
L131-132,“The Alpha Crombach coefficient was optimally estimated at the value of 97.” The correct expression for the Cronbach coefficient is “Cronbach's alpha”, which is generally between 0 and 1, not 97.
L 135, “Kruskall-Wallis” and “Mann-Whitney analysis” are two methods of analysis and should be stated separately with “and”. “Kruskall-Wallis” should be “Kruskal-Wallis”.
V. P3-8, Results section, it is better to pay attention to the normative and academic nature of data presentation (e.g., P5, p-value and R-value of L164-171, the decimal point should be “.” , not “,”; Table1 of P6-7, Table2 of P7-8 , it is better to summarize, categorize and compare stressors rather than itemize them; for Figure1 of P5, it is more appropriate to make a specific textual explanation of the data results. A large number of data results are not elaborated and interpreted in detail, and are scattered and fragmented; It is desirable to integrate and refine the data results to summarize and respond to the research objectives.
VI. P8-11, In Discussion section, the content is more diffuse, and a more structured statement is desirable.
The topics discussed jump around. In several places within the text, one sentence discusses burnout and the next discusses stress, anxiety, depression, sleep disorders, etc., which is inconsistent (e.g.L204-L216), and the authors could have added explanations of the connections between the topics.
L217-L228, The discussion of the results mentions the limitations of this study, and moreover exposes the lack of specification of this study, such as the representativeness of the sample and the recruitment of personnel (L217-L228).
Author Response
"Please see the attachment."

Reviewer 3 Report
Thank you for submitting the manuscript. I read your manuscript with great interest and attention, which deals with an important topic of great relevance.covid has been a really stressful event that has had significant consequences. However I would like you to stress more about what the situation of healthcare workers was before covid. I ask you this so as not to imply that the situation before the covid was idyllic.In this regard, I suggest you use the following references: doi: 10.11124/JBISRIR-2016-003309. doi: 10.3390/healthcare10081370. doi: 10.1097/GRF.0000000000000458.
Furthermore, I ask you to conform to the style of the Journal by providing: the code of the ethics committee in the appropriate section, the contribution of the authors, the funding, the conflict of interest and the acknowledgments. These sections before the references are mandatory.
Kind Regards
Author Response
"Please see the attachment."

Reviewer 4 Report
Please immediately improve your research manuscript according to the suggestions for improvement that I wrote in your manuscript!

Please immediately improve your research manuscript according to the suggestions for improvement that I wrote in your manuscript!
Author Response
"Please see the attachment."

Reviewer 5 Report
The topic of burnout in healthcare providers has been explored before, but the manuscript does provide a fresh perspective on some aspects (i.e., COVID-19). Consider expanding on highlighting current advancements to compact burnout and whether they work or not to provide readers with a richer background. All references cited are pertinent to the topics being discussed in the manuscript. While the research design gives a broad overview, it would benefit from a more structured and detailed approach, especially for the author-designed questionnaire items. The results are well-presented but the authors should consider reporting only statistically significant results in the tables. While the results support most of the conclusions drawn, some assertions could use more direct evidence or citations. I hope this review is helpful for the improvement and refinement of the manuscript.
The quality of written English is generally good. However, there are a few sections, especially in the introduction and discussion, where sentences could be rephrased for clarity and conciseness.
Author Response
"Please see the attachment."

Round 2
Reviewer 2 Report
Further amendments may be made to:
1. In the Abstract, "data methods" seems too detailed and can be appropriately reduced. Generally, the method and data in the Abstract section only need to explain the number of research objects, identity and data analysis method, without too much explanation, such as the informed consent of research objects, data collection methods, analysis software, etc. The Result statement would be better to further categorize some of the influencing factors (such as medical protective equipment, fear of infection, work schedule, etc.). In short, the overall content of the Abstract needs to be further condensed.
2. In the "Introduction" section, the author adds the concept of burnout, but it is better to further operationalize it, such as emotional performance and work performance. In addition, the content of literature review is supplemented (L76-L82; L93-L105; L127-L32), the relationship with this study can be explained.
3. In Table 1, participants are divided into two groups: COVID-19 (167) and non-COVID-19 (32) health care staff. The proportion of people in each demographic category should be the proportion of people in the group they belong to, not the proportion of total participants. The group comparison of demographic characteristics is to show that there are no significant difference between the two groups, which eliminates the influence of demographic characteristics on the target study variable. However, the authors did not explain it in result of Table 1. On the contrary, if there are significant differences in two variables (type of work, professional background), then the comparability of the two groups of participants will be questioned.
4. The author classified the contents of the scale according to the research objectives, but the score was not calculated according to the classification, and the two groups were still compared for each question in the scale.
5. There is too much content in Table 2, which seems not meet the research norms of general quantitative analysis. The author may needs to learn from other quantitative analysis models for further improvement. It is best to provide a score of burnout, such as descriptive statistics.
6. The author can further sort out the discussion section according to the updated results.
Author Response
"Please see the attachment."

Reviewer 4 Report
The article has been revised according to the suggestions for improvement that I gave, it's just that in writing the introductory sentence of the table, please add 2 sentences of introductory table, write the conclusion in more detail, the research design can be described in detail, not just one sentence, also include supporting references to the research design, and write questions study.
Author Response
"Please see the attachment."
